# The non-mitotic role of HMMR in regulating the localization of TPX2 and the dynamics of microtubules in neurons

Yi-Ju Chen[1], Shun-Cheng Tseng[2,3], Peng-Tzu Chen[1], Eric Hwang[1,3,4,5]*

[1]Institute of Molecular Medicine and Bioengineering, National Yang Ming Chiao Tung University, Hsinchu, Taiwan; [2]Department of Orthopedic Surgery, Changhua Christian Hospital, Changhua, Taiwan; [3]Department of Biological Science and Technology, National Yang Ming Chiao Tung University, Hsinchu, Taiwan; [4]Institute of Bioinformatics and Systems Biology, National Yang Ming Chiao Tung University, Hsinchu, Taiwan; [5]Center for Intelligent Drug Systems and Smart Bio-devices (IDS2B), National Yang Ming Chiao Tung University, Hsinchu, Taiwan

*For correspondence:
hwangeric@nycu.edu.tw

Competing interest: The authors declare that no competing interests exist.

**Abstract** A functional nervous system is built upon the proper morphogenesis of neurons to establish the intricate connection between them. The microtubule cytoskeleton is known to play various essential roles in this morphogenetic process. While many microtubule-associated proteins (MAPs) have been demonstrated to participate in neuronal morphogenesis, the function of many more remains to be determined. This study focuses on a MAP called HMMR in mice, which was originally identified as a hyaluronan binding protein and later found to possess microtubule and centrosome binding capacity. HMMR exhibits high abundance on neuronal microtubules and altering the level of HMMR significantly affects the morphology of neurons. Instead of confining to the centrosome(s) like cells in mitosis, HMMR localizes to microtubules along axons and dendrites. Furthermore, transiently expressing HMMR enhances the stability of neuronal microtubules and increases the formation frequency of growing microtubules along the neurites. HMMR regulates the microtubule localization of a non-centrosomal microtubule nucleator TPX2 along the neurite, offering an explanation for how HMMR contributes to the promotion of growing microtubules. This study sheds light on how cells utilize proteins involved in mitosis for non-mitotic functions.

## eLife assessment

In their **valuable** study, Chen et al. investigate the neuronal role of HMMR, a microtubule-associated protein typically associated with cell division. Their findings indicate that HMMR is necessary for proper neuronal morphology and the generation of polymerizing microtubules within neurites, potentially by promoting the function of TPX2. This **solid** body of work is the first step in deciphering the influence of a mitotic microtubule-associated protein in organizing microtubules in neurons and will be of interest to the neurobiology and cytoskeleton fields.

## Introduction

Animals interact with the environment through a highly intricate and organized network of interconnected cells. This network, known as the nervous system, is based on cells called neurons that are able to convey signals electrically and chemically. Neurons are also highly polarized with signal outputting compartments called axons that can extend over a meter in length and signal inputting compartments called dendrites that make the most elaborate tree branches pale in comparison. To

develop such a complex and polarized morphology, neurons go through a stereotypical morphogenetic process which was initially observed in vitro (*Dotti et al., 1988*). Like other cellular processes in which a polarized morphology is established and maintained, neuronal morphogenesis relies on the interplay between different cytoskeletons. The microtubule cytoskeleton in particular is involved in most if not all aspects of the neuronal morphogenetic process. Microtubules are dynamic biopolymers that can undergo polymerization and depolymerization through the addition and removal of α- and β-tubulin heterodimers (*Desai and Mitchison, 1997*). In addition to tubulin heterodimers that make up the bulk of microtubules, a collection of proteins called microtubule-associated proteins (MAPs) also regulate the dynamic nature of this cytoskeleton and play crucial roles in the morphogenesis of neurons (*Conde and Cáceres, 2009*; *Poulain and Sobel, 2010*). To better understand the role of MAPs in establishing and maintaining the neuronal morphology, we surveyed the microtubule-associated proteome in neurons using affinity purification and quantitative proteomics (Hwang et al., unpublished data). Surprisingly, hyaluronan-mediated motility receptor (HMMR), also known as receptor for hyaluronan mediated motility (RHAMM) or intracellular hyaluronic acid binding protein (IHABP) (*Hofmann et al., 1998*), is among the most abundant MAPs on neuronal microtubules. HMMR was originally identified as a hyaluronan-binding protein from the murine fibroblast (*Turley et al., 1987*). HMMR-targeting antibodies have since been used to demonstrate the involvement of surface-localized HMMR in hyaluronan-dependent motility in a variety of cell types (*Hardwick et al., 1992*; *Pilarski et al., 1993*; *Samuel et al., 1993*; *Savani et al., 1995*). More recent data demonstrate that HMMR is also an intracellular protein and interacts with the microtubule and actin cytoskeletons (*Assmann et al., 1999*; *Assmann et al., 1998*). In agreement with its role as a MAP, HMMR has also been shown to interact with other MAPs such as the microtubule motor dynein (*Maxwell et al., 2003*) and the microtubule nucleator TPX2 (*Chen et al., 2014*; *Groen et al., 2004*; *Scrofani et al., 2015*). Additionally, HMMR plays crucial roles in microtubule assembly near the chromosomes and at the spindle poles, spindle architecture, as well as mitotic progression (*Chen et al., 2014*; *Groen et al., 2004*; *Maxwell et al., 2003*; *Scrofani et al., 2015*). While the association of HMMR with microtubules is well documented in mitotic cells, very little is known about this interaction in non-mitotic cells. *Hmmr* mRNA has been shown to highly expressed in the proliferative regions of the nervous system in both developing amphibian and murine embryos (*Casini et al., 2010*; *Li et al., 2017*). The presence of HMMR protein in the adult brain and in dissociated primary neurons have also been documented (*Lindwall et al., 2013*; *Nagy et al., 1995*). Furthermore, HMMR is essential for proper mitotic spindle orientation in neural progenitor cells and the appropriate formation of various brain structures (*Connell et al., 2017*; *Li et al., 2017*). These observations demonstrate a crucial mitosis function of HMMR in neural progenitor cells. However, the function of HMMR extends beyond mitosis in neurons. Using antibodies targeting the cell surface HMMR or peptides mimicking hyaluronan binding domain of HMMR, it has been shown that HMMR is involved in neurite extension in primary neurons and intraocular brainstem transplants (*Nagy et al., 1995*; *Nagy et al., 1998*). These findings indicate that HMMR plays important roles in the nervous tissue and in non-mitotic neurons. However, whether HMMR exerts any effect on the microtubule cytoskeleton in neurons remains unexplored.

In this study, the function of HMMR in non-mitotic neurons is examined without any preconception regarding its localization. Using the shRNA-mediated depletion, HMMR knockdown negatively impacts the morphology of primary neurons. These morphological phenotypes include the decrease of axon and dendrite length as well as the reduction of axon branching complexity. The opposite phenotypes are observed in neurons transiently expressing *Hmmr*. Both endogenous and exogenous HMMR localizes to the microtubule cytoskeleton and exhibit punctate distribution along the neurites. In addition to its microtubule localization, HMMR is observed to enhance the stability and promote the formation of neuronal microtubules. We also found that the effect of HMMR on microtubule formation is due to its role in recruiting the microtubule nucleator TPX2 onto the microtubules. This work demonstrates that HMMR regulates the dynamics of microtubules independent of its mitotic function or its role on the centrosome in non-mitotic cells.

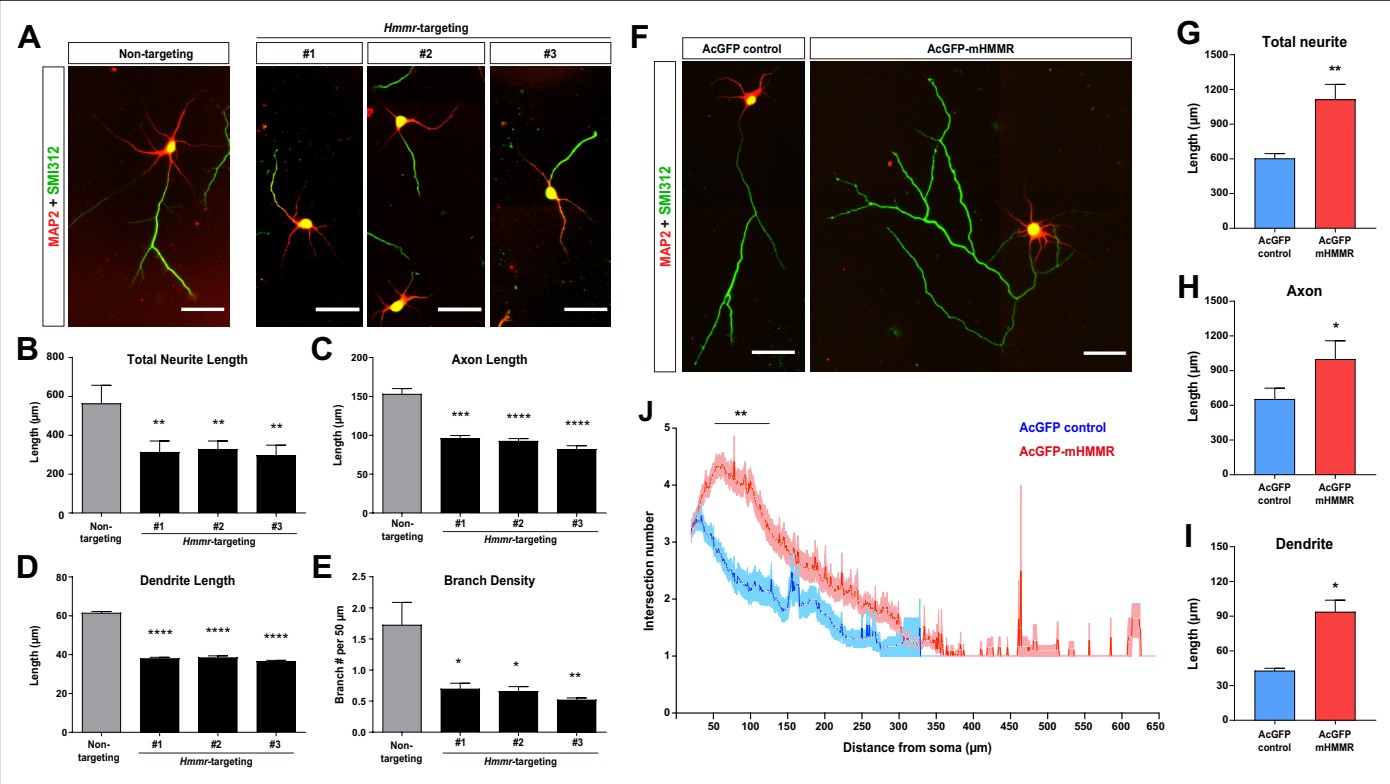

**Figure 1.** Hyaluronan-mediated motility receptor (HMMR) promotes neuronal morphogenesis. (**A**) Representative images of hippocampal neurons co-transfected with the EGFP-expressing and the indicated shRNA-expressing plasmids on 0 DIV and fixed on 4 DIV. Neurons were immunofluorescence stained with the dendrite marker MAP2 and the axon marker SMI312 (top). Quantification of (**B**) total neurite length per neuron, (**C**) axon length, (**D**) dendrite length, and (**E**) axon branch density (i.e. branch number per 50 µm of axon). *p<0.05, **p<0.01, ***p<0.001, ****p<0.0001, one-way ANOVA followed by Dunnett's post-hoc test. More than 20 neurons were analyzed per condition per repeat. (**F**) Representative images of hippocampal neurons transfected with AcGFP- or AcGFP-mHMMR-expressing plasmid on 0 DIV and fixed on 3 DIV. Neurons were immunofluorescence stained with the dendrite marker MAP2 and the axon marker SMI312. Quantification of (**G**) total neurite length per neuron, (**H**) axon length per neuron, and (**I**) dendrite length per neuron. *p<0.05, **p<0.01, two-tailed Student's t-test. More than 50 neurons were analyzed per condition per repeat. (**J**) Sholl analysis of the axon branching complexity. **p<0.01, two-way ANOVA followed by Sidak's post-hoc tests. The solid line and shaded area indicate mean and SEM collected from three independent repetitions (more than 50 neurons were analyzed per condition per repetition). All scale bars present 50 µm and all bar graphs are expressed as mean ± SEM from three independent repetitions.

The online version of this article includes the following figure supplement(s) for figure 1:

**Figure supplement 1.** Overexpressing human hyaluronan-mediated motility receptor (HMMR) rescues the effect of HMMR depletion in mouse neurons.

## Results

### HMMR regulates neuronal morphogenesis

To examine the role of HMMR in non-mitotic neurons, *Hmmr*-targeting shRNA was utilized to knock down *Hmmr* in mouse hippocampal neurons. Hippocampal neurons were selected because they exhibit a high morphological homogeneity, it has been estimated that 85~90% of hippocampal neurons are pyramidal neurons (*Banker and Goslin, 1998*). Three different shRNA sequences were used for depleting *Hmmr* in mouse neurons (*Figure 1—figure supplement 1A*). Upon HMMR depletion, a significant decrease in total neurite length, axon length, dendrite length, and axon branch density can be detected in dissociated hippocampal neurons (*Figure 1A–E*). To eliminate the possibility of the off-target effect, we performed the rescue experiment by co-transfecting plasmids expressing human HMMR (EGFP-hHMMR) and *Hmmr*-targeting shRNA into hippocampal neurons at 0 days in vitro (DIV) and incubated for 4 days before fixation and immunofluorescence staining. The expression of EGFP-hHMMR rescues the phenotype of HMMR knockdown (in both total neurite length and axon branching density) (*Figure 1—figure supplement 1B–E*).

In addition to the loss-of-function assay, we also performed the gain-of-function assay by transiently overexpressing a mouse HMMR fused to AcGFP1 (AcGFP-mHMMR) in hippocampal neurons.

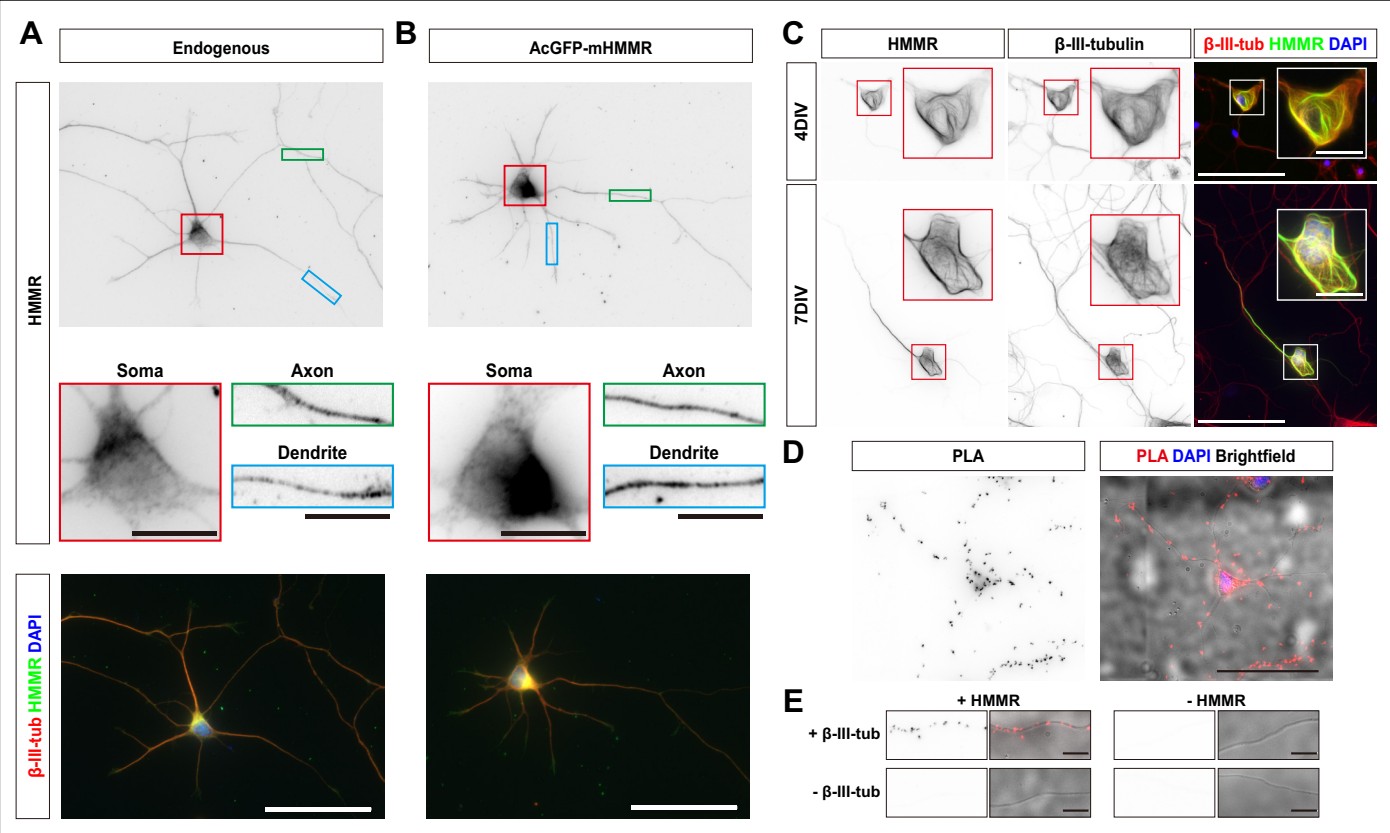

**Figure 2.** Hyaluronan-mediated motility receptor (HMMR) localizes to the microtubules in neurons. (**A**) Representative images of 3 DIV mouse hippocampal neurons immunofluorescence stained with antibodies against HMMR (green) and β-III-tubulin (red). Nuclei are visualized using DAPI (blue). HMMR images were inverted to improve visualization. (**B**) Representative images of 3 DIV mouse hippocampal neurons expressing AcGFP-mHMMR. Neurons were fixed and immunofluorescence stained with the antibody against β-III-tubulin (red). Nuclei are visualized using DAPI (blue). AcGFP-mHMMR images were inverted to improve visualization. Colored boxes indicate the magnified regions. The scale bars represent 10 μm and 50 μm in the colored boxes and the merged images, respectively. More than 50 neurons were observed for each condition, and HMMR exhibits similar localization in all neurons. (**C**) Representative images of 4 DIV (top) and 7 DIV (bottom) hippocampal neurons expressing AcGFP-mHMMR. Neurons were immunofluorescence stained with the β-III-tubulin antibody. AcGFP-mHMMR and β-III-tubulin signals were inverted to improve visualization. Red and white boxes at the soma are magnified in the insets. All images have the same scale and the scale bars present 50 μm. (**D**) Representative images of proximity ligation assay (PLA) on HMMR and β-III-tubulin in 3 DIV hippocampal neurons. The PLA image was inverted to improve visualization (left). DAPI was used to visualize the nuclei and brightfield microscopy was used to visualize the general appearance of neurons in the merged image (right). The scale bar presents 50 μm. (**E**) PLA puncta were present along the neurite shaft only when antibodies against HMMR and β-III-tubulin were both present. All images have the same scale and the scale bars represent 10 μm.

The online version of this article includes the following figure supplement(s) for figure 2:

**Figure supplement 1.** Validation of the hyaluronan-mediated motility receptor (HMMR) antibody.

**Figure supplement 2.** Transiently expressed AcGFP-mHMMR associates with microtubules in neurons.

**Figure supplement 3.** Hyaluronan-mediated motility receptor (HMMR) does not colocalize with microtubule plus-ends in neurons.

Consistent with the knockdown experiments, overexpressing HMMR results in the opposite phenotypes (i.e. an increase in total neurite length, axon length, dendrite length, and axon branch density) (*Figure 1F–J*). These functional analyses demonstrate that HMMR plays an important role in regulating the morphogenetic processes in non-mitotic neurons.

## HMMR is a microtubule-associated protein in neurons

To understand the cellular mechanism of HMMR in regulating neuronal morphogenesis, the localization of HMMR in neurons was examined. The HMMR antibody was first validated using shRNA-mediated HMMR depletion in neurons. A significant decrease in HMMR immunofluorescence signal was observed in the soma and along the neurites in HMMR-depleted neurons (*Figure 2—figure*

*supplement 1*), confirming the specificity of the HMMR antibody. Using this antibody, HMMR is detected along the entire neuron with higher abundance in the soma (*Figure 2A*). Upon careful examination, we found that both the endogenous HMMR and transiently expressed AcGFP-mHMMR shows punctate localization along the axon and dendrite (*Figure 2A–B*). Furthermore, when transiently overexpressed AcGFP-mHMMR reaches a high abundance level, it colocalizes with microtubules and sometimes causes the formation of looped microtubules in neurons (*Figure 2C*). Given that HMMR is known to interact with microtubules in mitotic cells (*Assmann et al., 1999*; *Maxwell et al., 2005*; *Tolg et al., 2010*), we examined whether the same interaction exists in neurons using the proximity ligation assay (PLA). We first examined whether the endogenous HMMR interacts with microtubules in neurons. Antibodies against HMMR and neuron-specific β-III-tubulin were used in this PLA. If the two primary antibodies are localized in close proximity (<40 nm), an enzymatic reaction will catalyze the amplification of a specific DNA sequence that can then be detected using a red fluorescent probe (*Söderberg et al., 2006*). Consistent with the idea that HMMR associates with neuronal microtubules, fluorescent PLA punta can be detected in the soma and along the neurite in 3 DIV hippocampal neurons (*Figure 2D*). These fluorescent PLA puncta can only be observed when both primary antibodies were present, indicating that these PLA signals are highly specific (*Figure 2E*). Furthermore, AcGFP-mHMMR-expressing plasmid was transfected into dissociated mouse hippocampal neurons at 0 DIV and incubated for 3 days before fixation and PLA. Antibodies against AcGFP and β-III-tubulin were selected for this PLA. Consistent with the result using endogenous HMMR, fluorescent PLA puncta can be observed in the soma and along the neurite in 3 DIV primary hippocampal neurons (*Figure 2—figure supplement 2A*). PLA signals can only be observed when both GFP and β-III-tubulin antibodies were present (*Figure 2—figure supplement 2B*). These results demonstrate that both endogenous and transiently expressed HMMR associate with microtubules in neurons.

Because the punctate distribution of HMMR on microtubules is reminiscent of microtubule plus-ends, we also examined the colocalization of HMMR and EB1 (a microtubule plus-end tracking protein) in neurons. Upon visual examination, HMMR and EB1 do not exhibit colocalization in neurons (*Figure 2—figure supplement 3A*). To quantify the extent of HMMR and EB1 colocalization in 1 DIV hippocampal neurons, linescans along the neurite and Pearson correlation coefficient were calculated. The average Pearson correlation coefficient is 0.28±0.25 (*Figure 2—figure supplement 3B*). This result indicates that the punctate HMMR localization in neurons does not represent microtubule plus-ends.

## HMMR stabilizes microtubules in neurons

The presence of HMMR on neuronal microtubules and the formation of looped microtubules in AcGFP-mHMMR overexpressing neurons (*Figure 3A*) suggests HMMR may be a microtubule-stabilizing factor. To test this possibility, the level of acetylated microtubules was quantified, as this post-translational modification is known to accumulate on stable and long-lived microtubules (*Schulze et al., 1987*). Consistent with our hypothesis, HMMR depletion via *Hmmr*-targeting shRNA produces a significant decrease in the level of acetylated microtubules in both axons and dendrites (*Figure 3B–D*). In contrast, the level of microtubule acetylation increases in both axons and dendrites of HMMR-overexpressing neurons (*Figure 3E–G*). Taken together, these data demonstrate that HMMR enhances microtubule stability in neurons.

In addition, we examined whether HMMR expression can resist the microtubule destabilizing effect of nocodazole in neurons. AcGFP or AcGFP-mHMMR expressing plasmid was introduced into dissociated hippocampal neurons on 0 DIV, incubated for 1 day to allow HMMR expression, and administered solvent (DMSO), 10 nM, 50 nM, or 100 nM nocodazole for two additional days before neurite length examination. While the addition of nocodazole causes a concentration-dependent reduction of total neurite length in both AcGFP and AcGFP-mHMMR expressing neurons, there are subtle differences in the susceptibility of neurite length to the concentration of nocodazole (*Figure 3H*). (1) 10 nM nocodazole treatment causes a significant reduction of neurite length in AcGFP expressing-neurons, but not in AcGFP-mHMMR-expressing neurons. This result indicates that AcGFP-mHMMR expression increases the tolerance of neurite elongation toward 10 nM nocodazole treatment. (2) 50 nM and 100 nM nocodazole treatment exhibits no statistical significance in AcGFP-expressing neurons, suggesting that 50 nM nocodazole has reached maximal effectiveness. In AcGFP-mHMMR expressing neurons, 100 nM nocodazole further reduces the neurite length compared to the 50 nM group. Taken

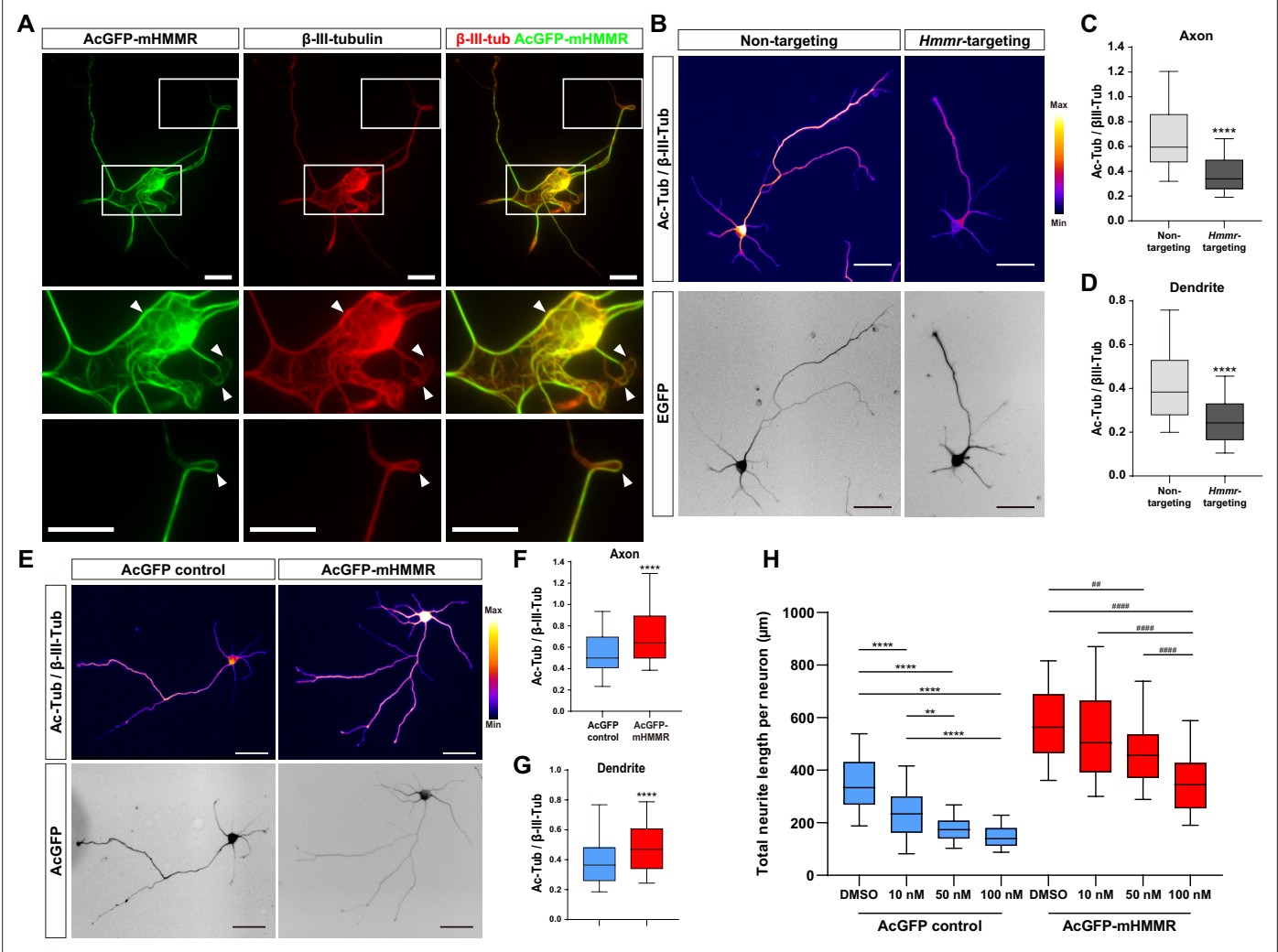

**Figure 3.** Hyaluronan-mediated motility receptor (HMMR) regulates microtubule stability in neurons. (**A**) Representative images of 4 DIV hippocampal neurons expressing AcGFP-mHMMR. Neurons were immunofluorescence stained with the antibody against β-III-tubulin. White boxes at the soma and the neurite tip are magnified. Arrowheads indicate looped microtubules. (**B**) Representative pseudo-colored acetylated-α-tubulin-to-β-III-tubulin ratio images of non-targeting shRNA (top left panel) or *Hmmr*-targeting shRNA (top right panel) expressing 4 DIV hippocampal neurons. The transfection indicator EGFP signal was inverted to improve visualization (bottom panels). Only neurons possessing both β-III-tubulin and EGFP signals were quantified. Quantification of the acetylated-α-tubulin-to-β-III-tubulin intensity ratio in axon (**C**) and dendrite (**D**). ****p<0.0001, two-tailed Mann-Whitney test. (**E**) Representative pseudo-colored acetylated-α-tubulin-to-β-III-tubulin ratio images of AcGFP-expressing control (top left panel) or AcGFP-mHMMR (top right panel) expressing 3 DIV hippocampal neurons. The AcGFP signal was inverted to improve visualization (bottom panels). Only neurons possessing both β-III-tubulin and AcGFP signals were quantified. Quantification of the acetylated-α-tubulin-to-β-III-tubulin intensity ratio within axon (**F**) and dendrite (**G**). **** *P*<0.0001, two-tailed Mann-Whitney test. (**H**) Quantification of total neurite length per neuron in AcGFP- or AcGFP-mHMMR expressing 3 DIV hippocampal neurons treated with or without the indicated concentration of nocodazole for 2 days. **p<0.01, **** *P*<0.0001, Kruskal-Wallis test followed by Dunn's post-hoc tests within the AcGFP expressing group. ##p<0.01, ####p<0.0001, Kruskal-Wallis test followed by Dunn's post-hoc tests within the AcGFP-mHMMR expressing group. All box plots are expressed as first quartile, median, and third quartile with whiskers extending to 5–95 percentile. More than 90 neurons were analyzed per condition per repeat. Scale bars represent 20 μm in (**A**) and 50 μm in (**B**) and (**E**).

together, these results are consistent with the idea that HMMR plays a microtubule stabilizing role in neurons.

## HMMR regulates neuronal microtubule dynamics

The association of HMMR with neuronal microtubules and its effect on microtubule stabilization suggest that HMMR may also be involved in the regulation of microtubule dynamics in neurons. To investigate this, we utilized the neuronal microtubule dynamics assay previously established in which EB3-mCherry is utilized as a fiduciary marker of growing microtubule plus-ends (***Chen et al.,***

*2017*). The rationale is that changes in microtubule dynamics can be detected as alterations in the velocity (or the speed of microtubule polymerization), persistence (the duration of time when EB3-mCherry comet can be followed), and/or frequency (the number of EB3-mCherry comets detected in a given time span). Plasmids expressing *Hmmr*-targeting shRNA and EB3-mCherry were introduced into dissociated neurons at 0 DIV and incubated for 4 days before fluorescence live cell imaging. To quantify microtubule dynamics, the neurite was separated into three different 10 µm regions: proximal, middle, and distal neurite (*Figure 4A*). These 3 regions were selected because of our previous publication (*Chen et al., 2017*), in which a significant reduction of EB3 frequency was detected at the tip and the base of the neurite but not in the middle of the neurite in the microtubule nucleator (TPX2) depleted neurons. The reason for this difference is due to the presence of GTP-bound Ran GTPase (RanGTP) at the tip and the base of the neurite. Since RanGTP has been shown to regulate the interaction between HMMR and TPX2 (*Scrofani et al., 2015*), it is possible that the same regulation mechanism exists in neurons. HMMR depletion results in a decrease of EB3-mCherry emanation frequency in all three neurite regions (*Figure 4B–C*). Moreover, a trend of increased microtubule polymerization velocity and a trend of decreased persistence are observed in the proximal neurite of HMMR-depleted neurons. Next, we examined whether opposite effects on microtubule dynamics can be detected in HMMR overexpressing neurons. Plasmids expressing AcGFP-mHMMR and EB3-mCherry were introduced into dissociated neurons at 0 DIV and incubated for 4 days before fluorescence live cell imaging and microtubule dynamics assay. Congruent with the depletion results, a significant increase in EB3-mCherry emanation frequency is observed at the proximal, middle, and distal neurites in HMMR-expressing neurons (*Figure 4D–E*). Further agreement comes from the decrease in microtubule polymerization velocity and the increase of microtubule persistence in neurons overexpressing HMMR. Both the decrease in microtubule polymerization velocity and the increase in microtubule persistence are consistent with the observation that HMMR can enhance the stability of neuronal microtubules. These data demonstrate that HMMR regulates the dynamics of microtubules in neurons.

## HMMR promotes TPX2-microtubule interaction in axons and dendrites

While changes in microtubule polymerization velocity and persistence in the previous section can be explained by the microtubule stabilizing effect of HMMR, the alteration in microtubule emanation frequency cannot. One explanation is that HMMR influences the function of another microtubule dynamics regulator in neurons. It has been shown that HMMR interacts with the microtubule nucleator TPX2 in a cell cycle-dependent manner (*Maxwell et al., 2005*) and this interaction is required for concentrating TPX2 at the spindle poles (*Groen et al., 2004*). Furthermore, we have shown that microtubule-bound TPX2 localizes along the neurite and is responsible for non-centrosomal microtubule formation in neurons (*Chen et al., 2017*). Combining these two observations, we hypothesized that HMMR affects the localization of TPX2 along the neurite which in turn regulates the formation of neuronal microtubules. To examine this hypothesis, the localization of TPX2 along the neurite was examined in neurons with or without HMMR depletion. Since the abundance of TPX2 along the neurite is rather low (*Chen et al., 2017*), PLA was utilized to detect this localization of TPX2. PLA using antibodies against TPX2 and β-III-tubulin produces numerous puncta along the neurite (*Figure 5A*). This punctate distribution appears similar to that produced by HMMR and β-III-tubulin PLA (*Figure 2C*). Consistent with our hypothesis, a statistically significant increase in PLA inter-punctal distance (the distance between PLA puncta) is detected in both axons and dendrites of HMMR-depleted neurons (*Figure 5B–C*). This result indicates that HMMR depletion reduces the localization of TPX2 on neuronal microtubules. These fluorescent PLA puncta can only be observed when both primary antibodies were present, indicating that these PLA signals are highly specific (*Figure 5D*). Next, the effect of HMMR overexpression on TPX2 localization is examined. Plasmids expressing AcGFP-mHMMR were introduced into hippocampal neurons on 0 DIV and incubated for 7 days before fixation and PLA. In agreement with the depletion experiment, a significant decrease in PLA inter-punctal distance is observed in HMMR overexpressing axon and dendrite (*Figure 5E–G*). This result indicates that HMMR overexpression enhances the localization of TPX2 on neuronal microtubules. PLA puncta can only be observed when both primary antibodies were present (*Figure 5H*). Taken together, these results demonstrate that HMMR promotes the interaction between TPX2 and microtubules in neurons.

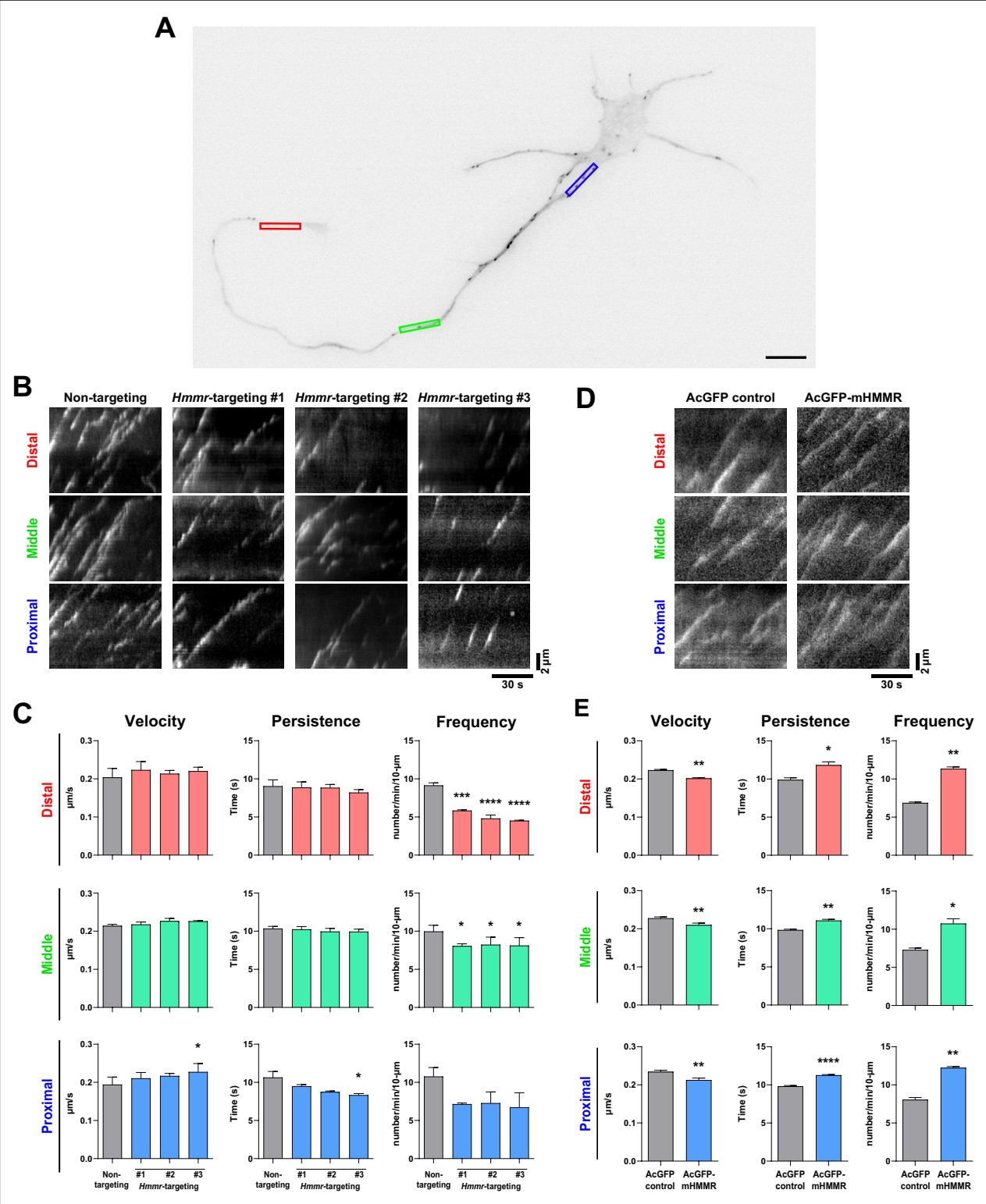

**Figure 4.** Hyaluronan-mediated motility receptor (HMMR) regulates the dynamics of neuronal microtubules. (**A**) Representative image of a 4 DIV EB3-mCherry-expressing cortical neuron. The color boxes indicate regions of quantification: red, green, and blue boxes represent the distal, middle, and proximal neurite, respectively. The scale bar presents 10 μm. (**B**) Representative kymographs of indicated neurons at different regions of the neurite. (**C**) Quantification of EB3-mCherry comets dynamics in B. *p<0.05, ***p<0.001, ****p<0.0001, one-way ANOVA followed by Dunnett's post-hoc tests. (**D**) Representative kymographs of indicated neurons at different regions of the neurite. (**E**) Quantification of EB3-mCherry comets dynamics in D. *p<0.05, **p<0.01, ****p<0.0001, two-tailed Student's t-test. At least 15 neurons were analyzed per condition per repeat. All bar graphs are expressed as mean ± SEM from three independent repeats.

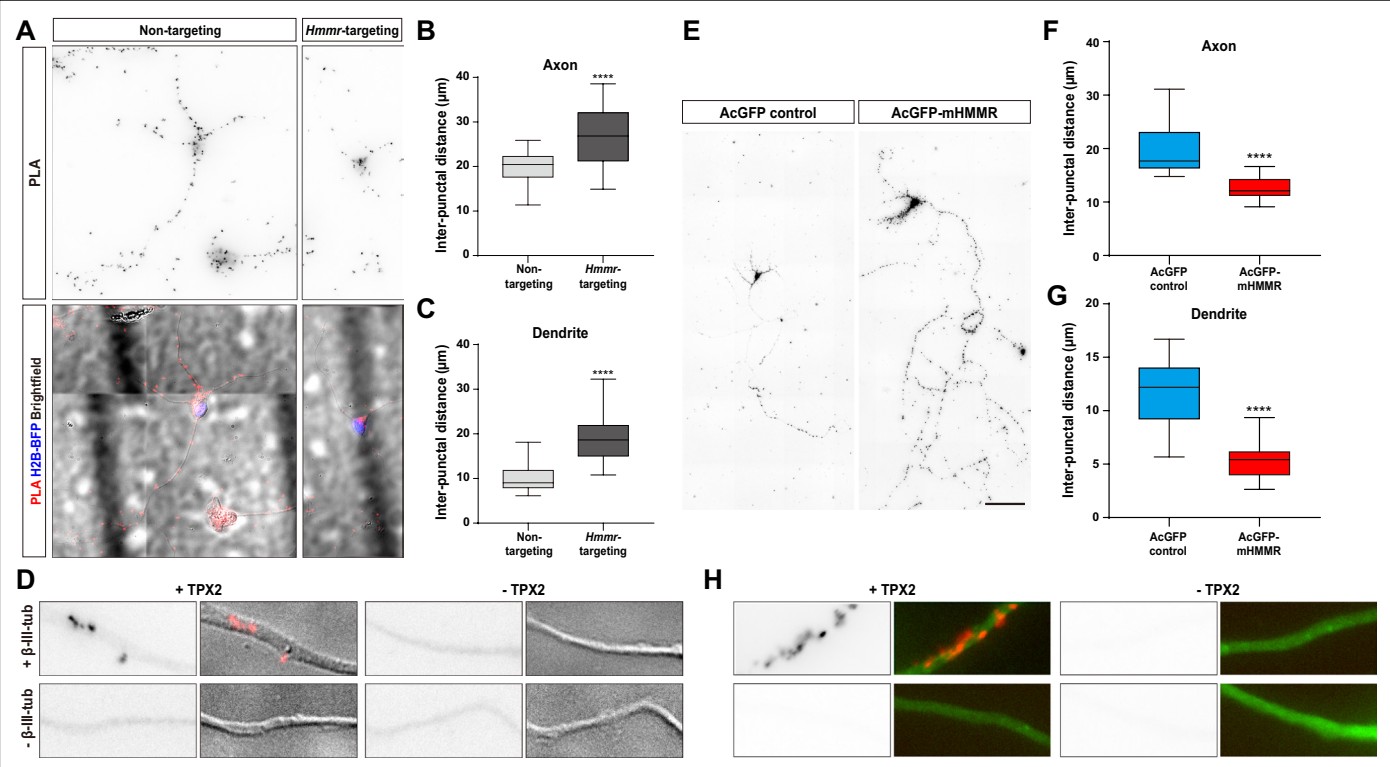

**Figure 5.** Hyaluronan-mediated motility receptor (HMMR) regulates the localization of TPX2 on microtubules in neurons. (A–D) Hippocampal neurons were co-transfected with indicated shRNA- and H2B-BFP-expressing plasmids at 0 DIV and cultured for 4 days before ice-cold methanol fixation. The *Hmmr*-targeting #1 shRNA was utilized to deplete HMMR. (A) Representative PLA images for TPX2 and β-III-tubulin (upper panel) and merged images of PLA, H2B-BFP, differential interference contrast (DIC) (lower panel) in 4 DIV hippocampal neurons. Quantification of the inter-punctal distance of PLA signals in axon (B) and dendrite (C). Only neurons possessing both PLA and H2B-BFP signals were quantified. ****p<0.0001, two-tailed Mann-Whitney test. (D) PLA signals were presented along the neurite only when both TPX2 and β-III-tubulin antibodies were present. (E–H) Hippocampal neurons were co-transfected with H2B-BFP- and either AcGFP- or AcGFP-mHMMR-expressing plasmids at 0 DIV and cultured for 7 days before ice-cold methanol fixation. (E) Representative PLA images for TPX2 and β-III-tubulin in 7 DIV hippocampal neurons. Quantification of the inter-punctal distance of PLA signals in axon (F) and dendrite (G). Only neurons possessing both PLA and H2B-BFP signals were quantified. ****p<0.0001, two-tailed Mann-Whitney test. All box plots are expressed as first quartile, median, and the third quartile with whiskers extending to 5–95 percentile. (H) PLA signals were presented along the neurite only when both TPX2 and β-III-tubulin antibodies were present. Dylight 488-conjugated secondary antibody was applied after PLA to stain β-III-tubulin antibody (shown in green). Scale bars represent 20 µm in (A), 5 µm in (D) (H), and 100 µm in (E).

## Discussion

In this study, we demonstrate that HMMR influences cellular morphogenesis in non-mitotic neurons using loss- and gain-of-function assays. Transient expression of HMMR promotes axon and dendrite elongation as well as enhances branch density, while depletion of HMMR produces the opposite phenotypes. Both endogenous and transiently expressed HMMR localize primarily to the microtubule cytoskeleton in neurons. The distribution of HMMR along the neurite has a punctate appearance but does not colocalize with the microtubule plus-ends. Using transient expression and shRNA-mediated depletion, it was discovered that HMMR enhances the stability and promotes the formation of neuronal microtubules. Finally, we show that HMMR regulates the recruitment of the microtubule nucleator TPX2 onto neuronal microtubules. These results demonstrate for the first time that HMMR plays an important role in regulating microtubules and morphogenesis in non-mitotic cells.

HMMR has been documented to be a hyaluronate-binding protein (***Turley et al., 1987***) as well as a microtubule-associated protein (***Assmann et al., 1999***). In more recent studies, HMMR is found to be associated with the centrosome in mitotic cells (***Maxwell et al., 2003***). It contains a central rod domain with coiled-coil structures flanked by two microtubule-binding domains at the N-terminus and a centrosome-targeting bZip motif at the C-terminus (***Assmann et al., 1999***; ***Maxwell et al., 2003***). While numerous studies focus on the role of HMMR in mitotic cells, studies on non-mitotic neurons are scarce. It has been documented that neutralizing HMMR with a functional blocking

antibody compromises neurite extension in cell lines and primary neurons (*Nagy et al., 1995*). A similar neutralizing strategy was used to demonstrate that HMMR is involved in axon outgrowth in an intraocular transplantation model (*Nagy et al., 1998*). Our loss-of-function assay is consistent with these aforementioned publications. It has been shown that the *Xenopus* HMMR homolog XRHAMM bundles microtubules in vitro (*Groen et al., 2004*). In addition, deleting proteins which promote microtubule bundling (e.g. doublecortin knockout, MAP1B/MAP2 double knockout) leads to impaired neurite outgrowth (*Bielas et al., 2007*; *Teng et al., 2001*). These observations are consistent with our data that overexpressing HMMR leads to the increased axon and dendrite outgrowth, while depleting it results in the opposite phenotype (*Figure 1*).

In addition, our data indicate that the increase in microtubule stability is the main mechanism driving the enhanced neurite outgrowth in neurons overexpressing HMMR (*Figure 3H*). It is worth noting that the elevated level of HMMR increases the branching density of axons (*Figure 1J*) and promotes the formation of looped microtubules (*Figure 3A*). This is consistent with the observations that looped microtubules are often detected in regions of axon branching site prior to branch formation (*Dent et al., 1999*; *Dent and Kalil, 2001*; *Purro et al., 2008*).

Using EB3-mCherry as a marker for growing microtubule plus-ends, we discovered that HMMR increases the amount of growing microtubules. These growing microtubules can either come from the de novo formation of microtubules or the repolymerization from paused or shrinking microtubules. The explanation for this increased number of growing microtubules lies in the observation that the localization of TPX2 (a branch microtubule nucleator) on microtubules is regulated by HMMR in neurons. This is consistent with previous publications showing that HMMR interacts with TPX2 and is required for concentrating TPX2 at the spindle pole (*Groen et al., 2004*; *Maxwell et al., 2005*; *Scrofani et al., 2015*). Given that GTP-bound Ran (RanGTP) promotes the interaction between HMMR and TPX2 in the cell-free system (*Scrofani et al., 2015*), it is tempting to hypothesize that RanGTP also regulates the HMMR-TPX2 interaction in neurons. It has previously been shown that cytoplasmic RanGTP promotes the formation of non-centrosomal microtubules at the neurite tip (*Huang et al., 2020*). This is due to the effect of RanGTP on releasing TPX2 from the inhibitory importin heterodimers (*Chen et al., 2017*). Our results suggest that the effect of cytoplasmic RanGTP on neuronal microtubules may come from recruiting TPX2 to the existing microtubules as well as enhancing its nucleator activity. It has been shown that compromising microtubule nucleation in neurons by SSNA1 mutant overexpression prevents proper axon branching (*Basnet et al., 2018*). Additionally, dendritic branching in *Drosophila* sensory neurons depends on the orientation of microtubule nucleation. Nucleation that results in an anterograde microtubule growth leads to increased branching, while nucleation that results in a retrograde microtubule growth leads to decreased branching (*Yalgin et al., 2015*). These results demonstrate the importance of microtubule nucleation on neurite branching. It is conceivable that overexpressing a microtubule nucleation promoting protein such as HMMR results in an increase in axon branching complexity.

Neurons are the communication units of the nervous system. The formation of their intricate shape is, therefore, crucial for the physiological function. Alterations in neuronal morphogenesis have a profound impact on how nerve cells communicate, leading to a variety of physiological consequences. These consequences conceivably include impaired neural circuit formation and function, compromised signal transmission between neurons, as well as altered anatomical structure of the CNS. Depending on the specific type and location of the morphogenetically altered neurons, the physiological consequences can include neurological disorders such as autism spectrum disorder (*Berkel et al., 2012*) and schizophrenia (*Goo et al., 2023*), as well as learning and memory deficits (*Winkle et al., 2016*). However, due to the involvement of HMMR in mitosis, most HMMR mutations are associated with familial cancers in humans (based on ClinVar data).

Given the importance of HMMR on spindle integrity and orientation, studies of HMMR on neural development have largely focused on these aspects. It has been documented that neural progenitor cells from *Hmmr* knockout or C-terminal truncation mice exhibit misoriented spindle and consequently these animals develop either megalencephaly or microcephaly (*Connell et al., 2017*; *Li et al., 2017*). Interestingly, *Hmmr* depletion in *Xenopus* showed defects in anterior neural tube closure (*Prager et al., 2017*). During CNS development, the anterior part of the neural tube closes and differentiates into the brain while the posterior part becomes the spinal cord. A series of cellular actions take place during the neural tube closure that includes polarization, migration, and intercalation (*Nikolopoulou*

*et al., 2017*). This failure in closing the anterior neural tube suggests an underlying defect in the microtubule cytoskeleton. Upon careful examination, interphase cells from the deep neural layer in *Hmmr* depleted embryos adopted a web-like microtubule organization instead of their typical linear microtubule organization. This data is consistent with our observation that HMMR plays a role in organizing the microtubule cytoskeleton in non-mitotic cells of the neural tissue. In support of this idea, *Hmmr* is amongst the highest expressed RNA in the corpus callosum relative to other tissues in adult humans (*Rouillard et al., 2016*). Because HMMR plays such a critical role in mitosis of neural progenitor cells, any loss-of-function *Hmmr* mutation in humans will likely result in embryonic lethality and prevent the observation of non-mitotic, microtubule-based phenotypes such as corpus callosum malformation or lissencephaly. It will be of great interest to examine the microtubule-rich brain structures such as corpus callosum or microtubule-dependent processes such as neuronal migration in *Hmmr* conditional knockout mice.

# Materials and methods

**Key resources table**

| Reagent type (species) or resource | Designation | Source or reference | Identifiers | Additional information |
|---|---|---|---|---|
| Gene (*M. musculus*) | *Hmmr* | GenBank | Gene ID: 15366 | |
| Transfected construct (*M. musculus*) | shRNA #1 | RNAi Consortium shRNA Library via RNAi Core of Academia Sinica | TRCN0000311803 | Lentiviral construct to express the *Hmmr*-targeting shRNA |
| Transfected construct (*M. musculus*) | shRNA #2 | RNAi Consortium shRNA Library via RNAi Core of Academia Sinica | TRCN0000311805 | Lentiviral construct to express the *Hmmr*-targeting shRNA |
| Transfected construct (*M. musculus*) | shRNA #3 | RNAi Consortium shRNA Library via RNAi Core of Academia Sinica | TRCN0000071592 | Lentiviral construct to express the *Hmmr*-targeting shRNA |
| Antibody | Anti-acetylated-α-tubulin (mouse monoclonal) | Abcam | ab24610 | IF (1:1000) |
| Antibody | Anti-β-III-tubulin (TUJ1) (mouse monoclonal) | BioLegend | 801202 | IF (1:4000) |
| Antibody | Anti-β-III-tubulin (TUBB3) (rabbit polyclonal) | BioLegend | 802001 | IF (1:2000) |
| Antibody | Anti-neurofilament (SMI312) (rabbit polyclonal) | BioLegend | 837904 | IF (1:1000) |
| Antibody | Anti-GFP (mouse monoclonal) | DSHB | 12A6 | IF (1:100) |
| Antibody | Anti-HMMR (E-19) (goat polyclonal) | Santa Cruz Biotechnology | sc-16170 | IF (1:50) |
| Antibody | Anti-MAP2 (rabbit polyclonal) | MilliporeSigma | AB5622 | IF (1:1000) |
| Antibody | Anti-TPX2 (rabbit polyclonal) | Oliver Gruss; *Gruss et al., 2002* | | IF (1:2000) |
| Recombinant DNA reagent | pCAG-AcGFP-mHMMR (plasmid) | This paper | | AcGFP-mHmmr expression vector |
| Recombinant DNA reagent | pEGFP-hHMMR (plasmid) | Christopher Maxwell; *Maxwell et al., 2003* | | |
| Commercial assay or kit | Duolink proximity ligation assay | Sigma-Aldrich | DUO92101 | |
| Software, algorithm | Prism | GraphPad v8.4.3 | RRID: SCR_002798 | |
| Software, algorithm | Fiji | Fiji | RRID: SCR_002285 | |
| Software, algorithm | NIS-Elements | Nikon | RRID: SCR_014329 | |

## Antibodies and reagents

Acetylated-α-tubulin antibody (ab24610) was purchased from Abcam (Cambridge, United Kingdom). β-III-tubulin antibodies TUJ1 and TUBB3 (801202 and 802001) as well as neurofilament monoclonal antibody SMI312 (837904) were from BioLegend (San Diego, CA). GFP antibody (12A6) was from DSHB (Iowa City, IA). HMMR antibody E-19 (sc-16170) was from Santa Cruz Biotechnology (Dallas, TX). MAP2 antibody (AB5622) and Duolink proximity ligation assay were from MilliporeSigma (Burlington, MA). TPX2 antibody was a kind gift from Oliver Gruss (*Gruss et al., 2002*). Alexa Fluor-conjugated secondary antibodies were from Thermo Fisher Scientific (Waltham, MA).

## Plasmids

The mouse *Hmmr*-expressing plasmid pCAG-AcGFP-mHMMR was cloned by inserting wild-type mouse *Hmmr* gene obtained from the mouse embryonal carcinoma P19 cell cDNA using PCR primers (5'-ATAGTCGACAGGCGTCAGAATGTCCTTTCCT-3' and 5'- TACCCGGGACTTCCATGATTCTTG AAGTTGCA-3') into pCAG-AcGFP-C3 using SalI and XmaI restriction endonucleases. The *Hmmr* gene obtained is 2385 bp in length and translates into a protein ~92 kDa in molecular weight. The human *HMMR*-expressing plasmid pEGFP-hHMMR was a kind gift from Dr. Christopher Maxwell (*Maxwell et al., 2003*). The mouse *Hmmr*-targeting shRNA plasmids were obtained from the RNAi Core of Academia Sinica (Taipei, Taiwan). The targeting sequences are 5'-GCCAGCTACTTGAAACAGAA A-3'(#1), 5'-CAGGCATTGTTGAATGAACAT-3' (#2), and 5'-GACTCTCAGAAGAATGATAAA-3' (#3).

## Neuron culture and transfection

All animal experimental procedures were approved by the Institutional Animal Care and Use Committee (IACUC) and in accordance with the Guide for the Care and Use of Laboratory Animals of National Yang Ming Chiao Tung University (approval reference number: NCTU-IACUC-110045). Dissociated hippocampal and cortical neuron cultures were prepared as previously described (*Chen et al., 2017*) with the following modifications. Hippocampi or cortexes from E17.5 mouse embryos were dissected, digested with trypsin-EDTA, and triturated. Dissociated neurons were seeded onto poly-L-lysine-coated coverslips ($2.5×10^3$ cells/cm$^2$ for low-density cultures and $3×10^4$ cells/cm$^2$ for regular-density cultures). Plasmids were introduced into neurons using Nucleofector II (Lonza, Basel, Switzerland) immediately before seeding or using Lipofectamine 2000 (Thermo Fisher Scientific) at the indicated number of days in vitro. Lipofectamine transfected cells were incubated for 4 hr and the medium containing the transfection mixture was then replaced with cortical neuron-conditioned neurobasal medium (Thermo Fisher Scientific, 21103049) (low-density cultures) or fresh neurobasal medium plus B27 supplement (regular-density culture).

## Indirect immunofluorescence staining

Cells on coverslips were fixed with 3.7% formaldehyde for 15 min at 37°C and then washed three times with PBS. Fixed cells were permeabilized with 0.25% triton X-100 in PBS for 5 min at room temperature or extracted in –20°C methanol for 10 min. For experiments that required cytosolic pre-extraction, cells on coverslips were permeabilized in 0.1% triton X-100 in PIPES buffer (0.1 M PIPES pH 6.9, 1 mM MgCl$_2$, and 1 mM EGTA) for 15 s, washed once with PIPES buffer, and fixed with 3.7% formaldehyde in PIPES buffer at 37°C for 30 min and then washed with PBS three times. Cells were then blocked with 10% BSA in PBS for 30 min at 37°C, incubated for 1 hr at 37°C with different primary antibodies: GFP (1:100), HMMR (1:50), MAP2 (1:1000), SMI312 (1:1000), TPX2 (1:2000), acetylated-α-tubulin (1:1000), TUBB3 (1:2000), and TUJ1 (1:4000). After primary antibody incubation, cells were washed with PBS three times and incubated with AlexaFluor-conjugated secondary antibodies (1:1000). All antibodies were diluted in 2% BSA in PBS. Coverslips with cells were washed with PBS three times and mounted with Fluoromount onto glass slides.

## In situ proximity ligation assay (PLA)

Cells were fixed in –20°C methanol for 10 min, washed with PBS, and then blocked in a chamber with Duolink II Blocking Solution for 30 min at 37°C. Primary antibodies used for different experiments were diluted in PBS containing 2% BSA at aforementioned dilutions and incubated for 1 hr at 37°C. Cells were then incubated with PLA probes diluted in Antibody Diluent for 1 hr at 37°C. Subsequent procedures were conducted according to the manufacturer's instructions.

## Microscopy acquisition

Fluorescence images were acquired on a Nikon Eclipse-Ti inverted microscope equipped with a Photometrics CoolSNAP HQ2 CCD camera, an Intensilight epi-fluorescence light source, and Nikon NIS-Element imaging software. 20×0.75 N.A. or 60×1.49 N.A. Plan Apochromat objective lenses were used to collect fluorescence images.

Live cell imaging was performed on a Nikon Eclipse-Ti inverted microscope equipped with a TIRF illuminator and a Tokai Hit TIZHB live cell chamber. Images were acquired using a 60×1.49 N.A. Plan Apochromat objective lens, a 561 nm DPSS laser, a Photometrics CoolSNAP HQ2 camera, and Nikon NIS-Elements imaging software. The built-in perfect focus system (PFS) was activated to maintain the axial position. Images were acquired every 500 milliseconds over a 2 min period. Only the neurons with clear EB3 comets were imaged.

## Image analysis

For neurite length analysis, fluorescence images were manually traced with the ImageJ plugin NeuronJ 1.4.1 (*Meijering et al., 2004*). Only neurons expressing both the transfection indicator (e.g. EGFP) and specific markers (e.g. β-III-tubulin, MAP2, SMI312) were analyzed. Only neurites longer than its soma diameter were analyzed.

For axon branching analysis, neurites were manually traced using the Fiji plug-in Simple Neurite Tracer (*Longair et al., 2011*) before being processed with the Fiji plug-in Sholl analysis (*Ferreira et al., 2014*).

For acetylated microtubule quantification, manually generated linescans along the neurite were used to obtain the signal of acetylated-α-tubulin and β-III-tubulin. The ratio of acetylated-α-tubulin/β-III-tubulin was calculated to represent the level of microtubule acetylation.

For microtubule plus-end dynamics analysis, NIS-Elements software was used to generate the kymograph for the EB3-mCherry images. All kymographs were generated using a window 10 μm in length and seven pixels in width. For proximal neurite analysis, the kymograph window started from the edge of the soma and extended outwards. For mid-neurite analysis, the kymograph window was centered at the midpoint of the neurite. For distal neurite analysis, the kymograph window started at the wrist of the growth cone and extended inwards. The speed and persistence time of EB3-mCherry were quantified from the kymograph by drawing a line along an EB3-mCherry event. Only EB3-mCherry movements that could be followed clearly for equal or more than four frames (1.5 s) were defined as an event. The emanating frequency of EB3-mCherry was quantified from the kymograph by counting the number of EB3-mCherry events per minute.

## Statistical analysis

All statistical analyses were performed using GraphPad Prism 8. Significant differences between the means were calculated with the indicated statistical methods.

## Additional information

### Funding

| Funder | Grant reference number | Author |
| --- | --- | --- |
| National Science and Technology Council | NSTC 111-2320-B-A49-015-MY3 | Eric Hwang |
| Ministry of Education | Center for Intelligent Drug Systems and Smart Bio-devices (IDS2B) | Eric Hwang |

The funders had no role in study design, data collection and interpretation, or the decision to submit the work for publication.

### Author contributions

Yi-Ju Chen, Data curation, Formal analysis, Validation, Investigation, Visualization, Writing - original draft; Shun-Cheng Tseng, Investigation, Writing - original draft; Peng-Tzu Chen, Data curation,

Investigation; Eric Hwang, Conceptualization, Resources, Supervision, Funding acquisition, Writing - review and editing

### Author ORCIDs
Yi-Ju Chen ⓘ http://orcid.org/0000-0001-9171-7186
Shun-Cheng Tseng ⓘ http://orcid.org/0000-0003-1564-6659
Peng-Tzu Chen ⓘ http://orcid.org/0009-0007-9172-1485
Eric Hwang ⓘ http://orcid.org/0000-0002-5188-581X

### Ethics
All animal experimental procedures were approved by the Institutional Animal Care and Use Committee (IACUC) and in strict accordance with the Guide for the Care and Use of Laboratory Animals of National Yang Ming Chiao Tung University.

Reviewer #1 (Public Review): https://doi.org/10.7554/eLife.94547.3.sa1
Reviewer #2 (Public Review): https://doi.org/10.7554/eLife.94547.3.sa2
Author response https://doi.org/10.7554/eLife.94547.3.sa3

---

# Additional files

### Supplementary files
• MDAR checklist

### Data availability
All data generated or analyzed during this study are available on DRYAD: https://doi.org/10.5061/dryad.cz8w9gjbz.

The following dataset was generated:

| Author(s) | Year | Dataset title | Dataset URL | Database and Identifier |
|---|---|---|---|---|
| Hwang E | 2024 | The non-mitotic role of HMMR in regulating the localization of TPX2 and the dynamics of microtubules in neurons | https://doi.org/10.5061/dryad.cz8w9gjbz | Dryad Digital Repository, 10.5061/dryad.cz8w9gjbz |

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
