## [Editor Report · eLife assessment]

In their **valuable** study, Chen et al. investigate the neuronal role of HMMR, a microtubule-associated protein typically associated with cell division. Their findings indicate that HMMR is necessary for proper neuronal morphology and the generation of polymerizing microtubules within neurites, potentially by promoting the function of TPX2. This **solid** body of work is the first step in deciphering the influence of a mitotic microtubule-associated protein in organizing microtubules in neurons and will be of interest to the neurobiology and cytoskeleton fields.

---

## [Referee Report · Reviewer #1 (Public Review)]

The microtubule cytoskeleton is essential for basic cell functions, enabling intracellular transport, and establishment of cell polarity and motility. Microtubule-associated proteins (MAPs) contribute to the regulation of microtubule dynamics and stability - mechanisms that are specifically important for the development and physiological function of neurons. Here, the authors aimed to elucidate the neuronal function of the MAP Hmmr, which they had previously identified in a (yet unpublished) quantitative study of the proteome associated with neuronal microtubules. The authors conduct well-controlled experiments to demonstrate the localization of endogenous as well as exogenous Hmmr on microtubules within the soma as well as all neurites of hippocampal neurons. Functional analysis using gain- and loss-of-function approaches demonstrates that Hmmr levels are crucial for neuronal morphogenesis, as the length of both dendrites and axons decreases upon loss of Hmmr and increases upon Hmmr overexpression. In addition to length alterations, the branching pattern of neurites changes with Hmmr levels. To uncover the mechanism of how Hmmr influences neuronal morphology, the authors follow the lead that Hmmr overexpression induces looped microtubules in the soma, indicative of an increase in microtubule stability. Microtubule acetylation indeed decreases and increases with Hmmr LOF and GOF, respectively. Together with a rescue of nocodazole-induced microtubule destabilization by Hmmr GOF, these results argue that Hmmr regulates microtubule stability. Highlighted by the altered movement of a plus-end-associated protein, Hmmr also has an effect on the dynamic nature of microtubules. The authors present evidence suggesting that the nucleation frequency of neuronal microtubules depends on Hmmr's ability to recruit the microtubule nucleator Tpx2. The authors discuss how branching may be regulated by Hmmr-mediated microtubule dynamics and speculate about the physiological significance of altered neuronal morphogenesis. Together, their work adds novel insight into MAP-mediated regulation of microtubules as a prerequisite for neuronal morphogenesis.

---

## [Referee Report · Reviewer #2 (Public Review)]

The mechanism of microtubule formation, stabilization, and organization in neurites is important for neuronal function. In this manuscript, the authors examine the phenotype of neurons following alteration in the level of the protein HMMR, a microtubule-associated protein with established roles in mitosis. Neurite morphology is measured as well as microtubule stability and dynamic parameters using standard assays. A binding partner of HMMR, TPX2, is localized. The results support a role for HMMR in microtubule stabilization in neurons.

The results show that HMMR is distributed as puncta on neurons using standard immunofluorescence and PLA. Depletion of HMMR reduced neurite length and extent of branching; reduced post-translational acetylation of neurite microtubules. Conversely, overexpression of HMMR increased resistance to nocodazole. The parameters of microtubule dynamics were also impacted by reduction or overexpression of HMMR. The authors discuss the possibility HMMR regulates neurite morphological changes via regulation of microtubule nucleation and dynamics.

---

## [Author Response]

The following is the authors’ response to the original reviews.

**eLife assessment**
In their valuable study, Chen et al. aim to define the neuronal role of HMMR, a microtubule-associated protein typically associated with cell division. Their findings suggest that HMMR is necessary for proper neuronal morphology and the generation of polymerizing microtubules within neurites, potentially by promoting the function of TPX2. While the study is recognized as a first step in deciphering the influence of HMMR on microtubule organization in neurons, reviewers note the current work has important gaps and would benefit from further exploration of the mechanism of microtubule stability by HMMR, the link between HMMR-mediated microtubule generation and morphogenesis, and the physiological implications of disrupting HMMR during neuronal morphogenesis.
**Public Reviews:**

**Reviewer #1 (Public Review):**
The microtubule cytoskeleton is essential for basic cell functions, enabling intracellular transport, and establishment of cell polarity and motility. Microtubule-associated proteins (MAPs) contribute to the regulation of microtubule dynamics and stability - mechanisms that are specifically important for the development and physiological function of neurons. Here, the authors aimed to elucidate the neuronal function of the MAP Hmmr, which they had previously identified in a quantitative study of the proteome associated with neuronal microtubules.The authors conduct well-controlled experiments to demonstrate the localization of endogenous as well as exogenous Hmmr on microtubules within the soma as well as all neurites of hippocampal neurons. Functional analysis using gain- and loss-of-function approaches demonstrates that Hmmr levels are crucial for neuronal morphogenesis, as the length of both dendrites and axons decreases upon loss of Hmmr and increases upon Hmmr overexpression. In addition to length alterations, the branching pattern of neurites changes with Hmmr levels. To uncover the mechanism of how Hmmr influences neuronal morphology, the authors follow the lead that Hmmr overexpression induces looped microtubules in the soma, indicative of an increase in microtubule stability. Microtubule acetylation indeed decreases and increases with Hmmr LOF and GOF, respectively. Together with a rescue of nocodazole-induced microtubule destabilization by Hmmr GOF, these results argue that Hmmr regulates microtubule stability. Highlighted by the altered movement of a plus-end-associated protein, Hmmr also has an effect on the dynamic nature of microtubules. The authors present evidence suggesting that the nucleation frequency of neuronal microtubules depends on Hmmr's ability to recruit the microtubule nucleator Tpx2. Together, these data add novel insight into MAP-mediated regulation of microtubules as a prerequisite for neuronal morphogenesis. While the data shown support the author's conclusions, the study also has several weaknesses:The study appears incomplete as the initial proteomics analysis which is referenced as an entry into the study is not presented. This surely is the authors' choice, however, without presenting this data set, it would make more sense if the authors first showed the localization of Hmmr on neuronal microtubules and then started with the functional analysis.

The reviewer suggests moving the Hmmr localization data in front of the loss- and gain-of-function data because we did not present the proteomics data. However, we still believe placing the loss- and gain-of-function data in the beginning is the better arrangement. This is because it allows the audience to see the drastic changes on neuronal morphology when HMMR is depleted or overly abundant. It also provides a better linkage between HMMR’s localization on microtubules and its effect on the stability and dynamics of microtubules.

Neurite branching is quantified, but the methods used are not consistent (normalized branch density vs. Sholl analysis) and there is no distinction between alterations of branching in dendrites vs. axons. This information should be added as it could prove informative with respect to the physiological function of Hmmr in neurite branching.

Sholl analysis is considered the gold standard in neurite branching analyses. However, in the knockdown experiment (Figure 1A~1E), HMMR-depleted neurons exhibited extremely short axons (<100 μm) and dendrites (<40 μm). Using Sholl analysis to assess the branching of these Hmmrdepleted neurons became unsuitable. That is why we used normalized branch density (Figure 1E) in the knockdown experiment and Sholl analysis (Figure 1J) in the overexpression experiment.

Regarding the branching difference between axons and dendrites, only axons exhibit branches at 4 DIV. Therefore, the branching analysis focuses on axons rather than on dendrites. We have revised the manuscript to clarify this.

The authors show that altered Hmmr levels affect neurite branching and identify an effect on microtubule stability and dynamics as a molecular mechanism. However, how branching correlates with or is regulated by Hmmr-mediated microtubule dynamics is neither addressed experimentally nor discussed by the authors.The physiological significance of altered neuronal morphogenesis also lacks discussion.

To discuss how branching correlates with or is regulated by HMMR-mediated microtubule dynamics, we have added the following paragraph into the Discussion section:

“It has been shown that compromising microtubule nucleation in neurons by SSNA1 mutant overexpression prevents proper axon branching (Basnet et al., 2018). Additionally, dendritic branching in *Drosophila* sensory neurons depends on the orientation of microtubule nucleation. Nucleation that results in an anterograde microtubule growth leads to increased branching, while nucleation that results in a retrograde microtubule growth leads to decreased branching (Yalgin et al., 2015). These results demonstrate the importance of microtubule nucleation on neurite branching. It is conceivable that overexpressing a microtubule nucleation promoting protein such as HMMR results in an increase of branching complexity.”

In terms of discussing the physiological significance of altered neuronal morphogenesis. We have added the following paragraph to the Discussion section:

“Neurons are the communication units of the nervous system. The formation of their intricate shape is therefore crucial for the physiological function. Alterations in neuronal morphogenesis have a profound impact on how nerve cells communicate, leading to a variety of physiological consequences. These consequences include impaired neural circuit formation and function, compromised signal transmission between neurons, as well as altered anatomical structure of the CNS. Depending on the specific type and location of the morphogenetically altered neurons, the physiological consequences can include neurological disorders such as autism spectrum disorder (Berkel et al., 2012) and schizophrenia (Goo et al., 2023), as well as learning and memory deficits (Winkle et al., 2016). However, due to the involvement of HMMR on mitosis, most HMMR mutations are associated with familial cancers (based on ClinVar data).”

Multiple times, the manuscript lacks a rationale for an experimental approach, choice of cell type, time points, regions of interest, etc. Also, a meaningful description of the methods and for how data were analyzed is missing, making the paper hard to read for someone not directly from the field.

We understand the reviewer’s comments regarding the lack of rationale for choosing the experimental approach, choice of cell type, time points, regions of interest, etc. As a result, we have added the rationales where appropriate to help readers from other fields to better understand the choice of cell type, time points, regions of interest, etc. A brief explanation is shown below:

Approach and timing: We employed both electroporation (immediate but milder expression) and lipofectamine transfection (delayed but stronger expression). We prioritized knocking down HMMR early in development, so electroporation was used. For overexpression experiments, we chose lipofectamine which allows high protein expression level to be achieved.Cell selection: Hippocampal neurons were chosen in experiments that involve morphological quantification due to their homogeneous morphology. On the other hand, cortical neurons were selected in experiments that require large amounts of neurons and/or experiments where we want to demonstrate the universality of a proposed hypothesis.Regions of interest (ROIs): In our previous publication (Chen et al., 2017), it was discovered that a significant reduction of EB3 emanation frequency can be detected at the tip and the base of the neurite but not in the middle of the neurite in TPX2-depleted neurons. The reason for this difference is due to the presence of GTP-bound Ran GTPase (RanGTP) at the tip and the base of the neurite. Since RanGTP has also been shown to regulate the interaction between HMMR and TPX2 in the cell-free system (Scrofani et al., 2015), it is possible that the same phenomenon can be observed in HMMR-depleted neurons. This is why we examined those 3 ROIs in Figure 4.

**Reviewer #2 (Public Review):**
The mechanism of microtubule formation, stabilization, and organization in neurites is important for neuronal function. In this manuscript, the authors examine the phenotype of neurons following alteration in the level of the protein HMMR, a microtubule-associated protein with established roles in mitosis. Neurite morphology is measured as well as microtubule stability and dynamic parameters using standard assays. A binding partner of HMMR, TPX2, is localized. The results support a role for HMMR in neurons.The work presented in this manuscript seeks to determine if a MAP called HMMR contributes to microtubule dynamics in neurons. Several steps, including validation of the RNAi, additional statistical analysis, use of cells at the same age in culture, and better documentation in figures, would increase the impact of the work.In many places, the data can be improved which might make the story more convincing. As presented, the results show that HMMR is distributed as puncta on neurons with data coming from a single HMMR antibody, and some background staining that was not discussed. In the discussion the authors state that HMMR impacts microtubule stability, which was evaluated by the presence of post-translational modification and resistance to nocodazole; the data are suggestive but not entirely convincing. The discussion also states that HMMR increases the “amount” of growing microtubules which was measured as the frequency of comet appearance. The authors did not comment on how the number of growing microtubules results in the observed morphological changes.

We actually tested several HMMR antibodies, including E-19 (Santa Cruz, sc-16170), EPR4054 (Abcam, ab124729), and a variety of antibodies provided by Prof. Eva Turley. E-19 performed the best in immunofluorescence (IF) staining and knockdown validation. The other antibodies either failed to detect HMMR in IF staining or generate excessive background signal. We understand that the final images are produced using a single antibody. But since we meticulous validated this antibody and that the localization of overexpressed HMMR is consistent with the endogenous HMMR, we are very confident about our data generated using this single antibody.

We have added the following paragraph in the Discussion section to elucidate how the number of growing microtubules result in the observed morphological changes such as an increase of axon branches:

“It has been shown that compromising microtubule nucleation in neurons by SSNA1 mutant overexpression prevents proper axon branching (Basnet et al., 2018). Additionally, dendritic branching in *Drosophila* sensory neurons depends on the orientation of microtubule nucleation. Nucleation that results in an anterograde microtubule growth leads to increased branching, while nucleation that results in a retrograde microtubule growth leads to decreased branching (Yalgin et al., 2015). These results demonstrate the importance of microtubule nucleation on neurite branching. It is conceivable that overexpressing a microtubule nucleation promoting protein such as HMMR results in an increase of branching complexity.

**Reviewer #1 (Recommendations for The Authors):**
(1) The manuscript jumps extensively between main figures and supplementary figures. Please check whether parts of the supplement could be moved to the main figures.

We understand the frustration of moving back and forth between the main figures and supplementary figures. After examining the manuscript, we decided to combine Figure 2A with Figure S3.

(2) In Figure 1, total neurite length between days 3 and 4 DIV does not appear to change - can this be true?Please check or else explain.

We carefully re-examined our raw data and found out the total neurite length of 4 DIV hippocampal neurons expressing non-targeting shRNA (Figure 1B) and that of 3 DIV hippocampal neurons expressing AcGFP (Figure 1G) are indeed very similar. The explanation is that the 3 DIV hippocampal neurons used for Figure 1G was cultured in low-density and in the presence of cortical neuron-conditioned neurobasal medium (as written in Methods, Neuron culture and transfection section). The low-density culture with minimal overlapping neurites allowed us to better quantify total neurite length, because neurons expressing AcGFP-mHMMR sprouted long and highly branched axons. However, the addition of cortical neuron-conditioned neurobasal medium promoted neurite elongation. This is the reason why the total neurite length of 4 DIV hippocampal neurons expressing non-targeting shRNA (Figure 1B) and that of 3 DIV hippocampal neurons expressing AcGFP (Figure 1G) is similar.

(3) Groen et al. have shown that Hmmr also bundles microtubules, a mechanism that surely is important for neuronal microtubules. Please discuss.

We thank the reviewer for pointing out that HMMR also bundles microtubules and have added this to our revised Discussion section:

“It has been shown that the *Xenopus* HMMR homolog XRHAMM bundles microtubules in vitro (Groen et al., 2004). In addition, deleting proteins which promote microtubule bundling (e.g., doublecortin knockout, MAP1B/MAP2 double knockout) leads to impaired neurite outgrowth (Bielas et al., 2007; Teng et al., 2001). These observations are consistent with our data that overexpressing HMMR leads to the increased axon and dendrite outgrowth, while depleting it results in the opposite phenotype (Figure 1).”

(4) Please explain why in Figure 4, cortical neurons were chosen for analysis and why and how the three different ROIs were picked.

To answer the question why we chose cortical neurons for the analyses in Figure 4, it will be important to explain why we used hippocampal neurons for other figures. Primary hippocampal neurons have a high homogeneity in terms of their morphology. This uniform morphology allows more consistent morphological quantification. Figure 4, however, does not involve morphological quantification. We are more confident to conclude that HMMR regulates microtubule dynamics if this effect can be detected in the relatively heterogeneous cortical neurons. These are the reasons why we chose to analyze cortical neurons in Figure 4.

In our previous publication (Chen et al., 2017), it was discovered that a significant reduction of EB3 emanation frequency can be detected at the tip and the base of the neurite but not in the middle of the neurite in TPX2-depleted neurons. The reason for this difference is due to the presence of GTP-bound Ran GTPase (RanGTP) at the tip of the neurite and in the soma. Since RanGTP has also been shown to regulate the interaction between HMMR and TPX2 in the cell-free system (Scrofani et al., 2015), it is possible that the same phenomenon can be observed in HMMR-depleted neurons. This was why we examined those 3 ROIs in Figure 4.

(5) Microtubule looping has been shown to occur in regions prior to branch formation (e.g. Dent et al. 2004). As the authors identify increased looping upon Hmmr GOF, this should be discussed.

We thank the reviewer for pointing out that microtubule looping occurs in regions of branch formation and have added this to our revised discussion:

“It is worth noting that the elevated level of HMMR increases the branching density of axons (Figure 1J) and promotes the formation of looped microtubules (Figure 3A). This is consistent with the observations that looped microtubules are often detected in regions of axon branch formation (Dent et al., 1999; Dent and Kalil, 2001; Purro et al., 2008).”

**Reviewer #2 (Recommendations for The Authors):**
(1) The work seeks to gain insight into microtubule behavior in neurons, an important issue.(2) Several steps, including validation of the RNAi, additional statistical analysis, use of cells at the same age in culture, and better documentation in figures, would increase the impact of the work.(3) Figure 1 documents the results of experiments in which the HMMR protein was depleted using shRNA. A western blot of cell extracts from control and depleted cells is needed to verify that the protein level is reduced; alternatively, documentation of the reduction in RNA levels in treated cells could be provided. Neurite, axon, and dendrite length and branch density are measured. The neurite length is in microns, and the axon length is normalized to 100% of the non-treated cells. Please use the same for measures for easier comparison. Looking at the images in Figure 1, the length of the dendrites does not look different in the examples shown, whereas the axon appears shorter. This impression is not supported by the quantification. Are representative images shown? Additionally, the authors should report the values for each replicate of the experiment and compare the three averages rather than comparison of lengths from all measurements. A related issue is that the dendrites do not look longer in panel F, following overexpression of HMMR. For examples of using averages of replicates see: https://pubmed.ncbi.nlm.nih.gov/32346721/

The reviewer mentioned that Western blot of cell extracts or RNA quantification from control and depleted cells are needed to verify that the protein level is reduced.

Unfortunately, these assays are extremely difficult to perform in primary neurons due to the low transfection efficiency. We believe that the consistent knockdown phenotype from 3 different shRNA sequences (Figure 1A-D) and the immunofluorescence staining in depleted primary neurons (Figure S2) are sufficient to confirm that HMMR level is reduced.

We revised Figure 1C, 1D, 1H, 1I so that axon and dendrite lengths are all in micron.

We selected another image for the non-targeting control in Figure 1A to better demonstrate the reduction of dendrite length when HMMR is knocked down.

We thank the reviewer for the suggestion of comparing the three average values rather than comparing all measurements. We have performed statistical analyses for all our data using the average values and revised the graphs accordingly. While the P-values changed, our conclusions remain the same.

We thank the reviewer for pointing out this discrepancy and have selected another image of the AcGFP control for Figure 1F to better demonstrate the increase of dendrite length when HMMR is overexpressed.

(4) Given the changes in neurite morphology, the authors examine the localization of endogenous and overexpressed. The supplemental figures (see S2 and S3) show evidence that HMMR is present in a punctate pattern by conventional immunofluorescence. This is reasonable evidence that the protein is in a linear pattern along cytoskeletal microtubules and that the signal is present in puncta. Please move this to the main text, perhaps replacing Figure 2A, which is low magnification and very hard to see the HMMR staining. Additionally, the level of overexpression of HMMR is not mentioned. Please address this; were cells with similar levels of overexpression selected? Did the result depend on the overexpression? A related issue is the DIV for the cells - some are examined earlier and some at later times; does this impact the results? Please provide information or perform experiments with consistent timing. For the immunofluorescence, were multiple antibodies tried to see if the result was the same with each? Were different fixations, in addition to methanol, utilized?

We have replaced Figure 2A with Figure S3 based on the reviewer’s suggestion.

In the HMMR overexpression experiments, we used HMMR antibody and immunofluorescence staining to confirm that the overexpression is achieved. However, we did not quantify to what extend HMMR was overexpressed.

We performed all the depletion experiments on 4 DIV to maximize knockdown efficiency and performed all the overexpression experiments on 3 DIV to prevent excessive axon fasciculation. Nonetheless, we examined the effect of HMMR depletion on neuronal morphology on 3 DIV. The trend of reduced total neurite length, axon length, and dendrite length can be observed, but no statistical significance can be detected. We also examined the effect of HMMR overexpression on neuronal morphology on 4 DIV and did observe an increase of total neurite length, axon length, and dendrite length. But the overlapping and bundled axons made reliable quantification extremely difficult.

We actually tested multiple HMMR antibodies, such as E-19 (Santa Cruz, sc-16170), EPR4054 (Abcam, ab124729), and a variety of antibodies provided by Prof. Eva Turley. E19 performed the best in immunofluorescence (IF) staining and knockdown validation. The other antibodies either failed to detect HMMR in IF staining or generate excessive background signal. We also tested various fixation methods, including 37°C formaldehyde fixation, -20°C methanol fixation, 37°C formaldehyde followed by -20°C methanol fixation. All fixation methods generated similar IF staining pattern using the E-19 antibody, but 3.7% formaldehyde fixation produced the highest signal.

(5) In Figure 2 C it is hard to see DAPI fluorescence. Are the white areas in the merge with bright cell nuclei? Is Figure 2C control or overexpressing cells? If this is endogenous, is there less signal in PLA compared with S4, which was in culture longer and is overexpressed prior to using PLA for detection?

The white areas in Figure 2C the reviewer mentioned are not cell nuclei, they are actually bubbles formed within the mounting medium.

HMMR detected in Figure 2C is endogenous. We did not quantitatively compare the PLA signals in Figure 2C and those in Figure S4. This is because the PLA signals in Figure 2C are generated using anti-HMMR (to detect endogenous HMMR) and anti-β-III-tubulin antibodies while those in Figure S4 are generated using anti-AcGFP (to detect overexpressed AcGFP-mHMMR) and anti-β-III-tubulin antibodies. Since the affinity of the two antibodies (i.e., anti-HMMR and anti-AcGFP) toward their antigens is different, comparing the PLA signals is not informative.

(6) The images of the endogenous HMMR (Fig S3) and the PLA with tubulin and HMMR antibodies are not the same (2C). The "dots" in PLA are widely separated; gauging from the marker bar length of 50 μm, the small clusters of dots are about 10 μm apart. In Figure S3, the puncta are much more closely spaced, appearing almost in a linear fashion along the microtubules. Enlarging the PLA image shows that each dot is very small - just a few pixels - please provide additional explanation including the minimal detection limit for the method, and why the images differ. If the standard immunofluorescence signal was enhanced, for example with the use of two secondaries, what is observed? Is the distribution of HMMR similar for both dendrites and axons? Microtubule polarity differs in these locations, so greater attention to this point seems of interest. There is a significant amount of punctate HMMR in the cytoplasm (or outside the cytoplasm?) in Figure S5; this is concerning. Please outline the cell edge for ease of visualization. What is the distribution of HMMR in a cell that has been treated with cold and/or nocodazole to disassemble the microtubules? is the signal lost?

The reasons images of the endogenous HMMR (Figure S3) and the PLA with tubulin and HMMR antibodies (Figure 2C) differ are due to the following reasons. o PLA utilizes two primary antibodies to target two different epitopes on HMMR and βIII-tubulin. It is conceivable that not every anti-HMMR antibody has the correct orientation and/or proximity (<40 nm) toward the anti-β-III-tubulin antibody to enable DNA amplification. This results in the shortage of PLA puncta compared to immunofluorescence signals.

The creator of PLA has pointed out that in situ PLA is a method based upon equilibrium reactions and several enzymatic steps. Therefore, only a fraction of the inter-acting molecules is detected (Weibrecht et al., 2010).

We have not used signal enhancing immunofluorescence staining methods [e.g., using tertiary antibodies or tyramide signal amplification (TSA)] to detect HMMR. This is mainly because HMMR signal is strong enough to be detected using standard immunofluorescence staining.

Regarding the question “Is the distribution of HMMR similar for both dendrites and axons?” The reviewer raised a very important issue about the polarity difference of microtubules in axons (uniform) and dendrites (mixed). We were aware of such issue and very carefully examined the distribution and signal intensity of HMMR in axons vs dendrites. However, no differences were detected.

The reviewer mentioned that “there is a significant amount of punctate HMMR in the cytoplasm (or outside the cytoplasm?) in Figure S5; this is concerning. Please outline the cell edge for ease of visualization.” Instead of outlining the cell edge, we have selected another image to facilitate the visualization of HMMR signals. There are indeed HMMR signals outside the cell. However, these outside signals are usually weaker and smaller in size compared to those inside the cell.

After the examination of neurons expressing AcGFP-mHMMR with or without 100 nM nocodazole treatment, we did not notice any difference of AcGFP-mHMMR in distribution. We did not examine the distribution and signal intensity of the endogenous HMMR.

(7) To determine if HMMR alters microtubule stability, the authors examine the distribution of acetylated tubulin and resistance to nocodazole-induced microtubule disassembly. In Figure 3 please show immunofluorescence images of the acetylated tubulin staining, not just the ratio images; the color is not obviously different in the various panels shown. For statistical analysis, see the comment above for Figure 1. For the nocodazole experiment, a similar change in neurite length following drug treatment was observed (Figure 3H), for the experimental and control, even though the starting length was greater in the overexpressing cells. Please consider the possibility that in both cases the microtubules are only partially resistant to nocodazole and that HMMR is not changing the fraction of microtubules that are sensitive to the drug. The cells were treated at 3 DIV; the authors note that more stable microtubules accumulate with time; how does time in culture impact stability? Often, acute treatment with a high concentration of nocodazole is used to assay microtubule stability; here the authors used a low (nM) concentration for 2 days (chronic). Why not use a higher concentration (1-10 μM) for a shorter incubation? The data show that overexpression of HMMR results in curved, buckled microtubules are these microtubules more acetylated and/or retained after nocodazole treatment?

The reviewer suggested that we show immunofluorescence images of the acetylated tubulin staining, not just the ratio images. But we still believe showing the ratio images is the better approach. This is because the microtubules density can be different from neuron to neuron. Showing acetylated tubulin may provide a false impression when the overall microtubule density is higher or lower in a particular neuron. We realized that “16 colors” pseudo-color scheme has the cyan color at the lower intensity which can sometimes be confused with the white color at the higher intensity. Therefore, we changed the pseudocolor from “16 colors” to “fire” for Figure 3B and 3E to better visualize these images so that they appear more consistent with the quantitative data.

The reviewer raised a very good question regarding the possibility that HMMR is not changing the fraction of microtubules that are sensitive to nocodazole. We re-conducted the same experiment and used a series of different nocodazole concentrations. While the addition of nocodazole causes a concentration-dependent reduction of total neurite length in both AcGFP and AcGFP-mHMMR expressing neurons, there are subtle differences in the susceptibility of neurite length to the concentration of nocodazole. (1) 10 nM nocodazole treatment causes a significant reduction of neurite length in AcGFP expressing neurons, but not in AcGFP-mHMMR expressing neurons. This result indicates that AcGFP-mHMMR expression increases the tolerance of neurite elongation toward 10 nM nocodazole treatment. (2) 50 nM and 100 nM nocodazole treatment exhibits no statistical significance in AcGFP expressing neurons, suggesting that 50 nM nocodazole has reached maximal effectiveness. In AcGFP-mHMMR expressing neurons, 100 nM nocodazole further reduces the neurite length compared to the 50 nM group. These results argue against the possibility that HMMR does not change the fraction of microtubules that are sensitive to nocodazole. We have revised Figure 3H accordingly.

The reviewer asked why we did not use the acute nocodazole treatment (μM concentration) to assess the effect of Hmmr on microtubule stability. This is because we used the neurite length as an indicator for microtubule stability. That is why the chronic treatment was chosen to produce a more detectable effect on neurite length.

The reviewer asked whether the looped microtubules caused by HMMR overexpression are more acetylated and/or nocodazole resistant. While we do not have direct evidence to answer the reviewer’s question, we can deduce the answer from our observations. We noticed that looped microtubules are only present when HMMR is highly expressed (i.e., using lipofection to introduce HMMR-expressing plasmid) but not when HMMR is mildly expressed (i.e., using electroporation to introduce HMMR-expressing plasmid). From these observations, we can conclude that HMMR is more abundantly present on looped microtubules. Since HMMR overexpression leads to higher microtubule acetylation (Figure 3E), looped microtubules which contains more HMMR are most likely to be more acetylated.

(8) An additional measure of microtubule dynamics is to measure the growth of microtubules using a live cell marker for microtubule plus ends. Such experiments were performed, using tagged EB3. The images are rather fuzzy. Parameters of microtubule dynamics were measured at three locations - is there data that the authors can cite about any differences in dynamics in control cells at these locations? They look very similar, so it is not clear why the different locations were used. It is not possible to learn much from the kymographs which look similar for all panels; I would remove these unless they can be changed or labeled to help the reader. Data is presented for three shRNA reagents. No data are presented to document the extent to which the protein is depleted with these reagents. This should be fixed. Alternatively, an RNAi pool could be utilized. Is there a control for off-target effects? For the analysis were all the comets used to generate the average values? What about a comparison of the average of each trial - not each comet?

In our previous publication (Chen et al., 2017), it was discovered that a significant reduction of EB3 emanation frequency can be detected at the tip and the base of the neurite but not in the middle of the neurite in TPX2-depleted neurons. The reason for this difference is due to the presence of RanGTP at the tip and the base of the neurite. Since RanGTP has also been shown to regulate the interaction between HMMR and TPX2 in the cell-free system (Scrofani et al., 2015), it is possible that the same phenomenon can be observed in HMMR-depleted neurons. This is why we examined those 3 ROIs in Figure 4.

We notice that photobleaching causes the EB3-mCherry signal to diminish at later time points, which made it difficult to observe the differences amongst kymographs. In the revised Figure 4B and 4D, we removed the second half of all the kymographs to make the differences more obvious.

The reviewer mentioned that there are no data documenting the extent to which the protein is depleted with the shRNAs. These data are shown in Figure S2, in which we quantified the HMMR protein level in the soma and along the neurite in neurons expressing different shRNA molecules.

The reviewer asked whether there is a control for off-target effects. The answer is yes. We performed the rescue experiment to control for off-target effects, which is shown in Figure S1.

We revised Figure 4 so that the dynamic properties of EB3 are quantified using the average of each experimental repetition.

(9) In a final experiment, the authors examine the distribution of TPX2, a binding partner of HMMR. Include a standard immunofluorescence in addition to PLA to illustrate the distribution of TPX2. The quantification used was the inter puncta distance; please quantify the signal in control and treated cells.

The reviewer asked us to include a standard immunofluorescence staining to illustrate the distribution of TPX2. We have done that in our previous publication (Chen et al., 2017) and TPX2 localizes primarily to the centrosome (https://www.nature.com/articles/srep42297/figures/2). In order to enhance the weak signal of TPX2 along the neurite, we actually needed to use PLA in that publication (https://www.nature.com/articles/srep42297/figures/3).

Proximity ligation assay (PLA) generates fluorescent signals based on a local enzymatic reaction which catalyzes the amplification of a specific DNA sequence that can then be detected using a red fluorescent probe. Because this enzymatic reaction is not linear, the amount of amplified DNA nor the intensity of the fluorescence does not correlate with the strength of the interaction (Soderberg et al., 2006). As a result, quantification of PLA is typically done by counting the number of fluorescent puncta per unit area or by calculating the area containing fluorescent signal (not signal intensity) per unit area in the case that PLA signals are too strong and coalesced. That is why our quantification is based on the distance between PLA fluorescent puncta, not the fluorescent signal intensity.

References

Basnet, N., H. Nedozralova, A.H. Crevenna, S. Bodakuntla, T. Schlichthaerle, M. Taschner, G. Cardone, C. Janke, R. Jungmann, M.M. Magiera, C. Biertumpfel, and N. Mizuno. 2018. Direct induction of microtubule branching by microtubule nucleation factor SSNA1. Nat. Cell Biol. 20:1172-1180.

Berkel, S., W. Tang, M. Trevino, M. Vogt, H.A. Obenhaus, P. Gass, S.W. Scherer, R. Sprengel, G. Schratt, and G.A. Rappold. 2012. Inherited and de novo SHANK2 variants associated with autism spectrum disorder impair neuronal morphogenesis and physiology. Hum. Mol. Genet. 21:344-357.

Bielas, S.L., F.F. Serneo, M. Chechlacz, T.J. Deerinck, G.A. Perkins, P.B. Allen, M.H. Ellisman, and J.G. Gleeson. 2007. Spinophilin facilitates dephosphorylation of doublecortin by PP1 to mediate microtubule bundling at the axonal wrist. Cell. 129:579-591.

Chen, W.S., Y.J. Chen, Y.A. Huang, B.Y. Hsieh, H.C. Chiu, P.Y. Kao, C.Y. Chao, and E. Hwang. 2017. Ran-dependent TPX2 activation promotes acentrosomal microtubule nucleation in neurons. Sci. Rep. 7:42297.

Dent, E.W., J.L. Callaway, G. Szebenyi, P.W. Baas, and K. Kalil. 1999. Reorganization and movement of microtubules in axonal growth cones and developing interstitial branches. J. Neurosci. 19:8894-8908.

Dent, E.W., and K. Kalil. 2001. Axon branching requires interactions between dynamic microtubules and actin filaments. J. Neurosci. 21:9757-9769.

Goo, B.S., D.J. Mun, S. Kim, T.T.M. Nhung, S.B. Lee, Y. Woo, S.J. Kim, B.K. Suh, S.J. Park, H.E. Lee, K. Park, H. Jang, J.C. Rah, K.J. Yoon, S.T. Baek, S.Y. Park, and S.K. Park. 2023. Schizophrenia-associated Mitotic Arrest Deficient-1 (MAD1) regulates the polarity of migrating neurons in the developing neocortex. Mol. Psychiatry. 28:856-870.

Groen, A.C., L.A. Cameron, M. Coughlin, D.T. Miyamoto, T.J. Mitchison, and R. Ohi. 2004. XRHAMM functions in ran-dependent microtubule nucleation and pole formation during anastral spindle assembly. Curr. Biol. 14:1801-1811.

Purro, S.A., L. Ciani, M. Hoyos-Flight, E. Stamatakou, E. Siomou, and P.C. Salinas. 2008. Wnt regulates axon behavior through changes in microtubule growth directionality: a new role for adenomatous polyposis coli. J. Neurosci. 28:8644-8654.

Scrofani, J., T. Sardon, S. Meunier, and I. Vernos. 2015. Microtubule nucleation in mitosis by a RanGTP-dependent protein complex. Curr. Biol. 25:131-140.

Soderberg, O., M. Gullberg, M. Jarvius, K. Ridderstrale, K.J. Leuchowius, J. Jarvius, K. Wester, P. Hydbring, F. Bahram, L.G. Larsson, and U. Landegren. 2006. Direct observation of individual endogenous protein complexes in situ by proximity ligation. Nat. Methods. 3:995-1000.

Teng, J., Y. Takei, A. Harada, T. Nakata, J. Chen, and N. Hirokawa. 2001. Synergistic effects of MAP2 and MAP1B knockout in neuronal migration, dendritic outgrowth, and microtubule organization. J. Cell Biol. 155:65-76.

Weibrecht, I., K.J. Leuchowius, C.M. Clausson, T. Conze, M. Jarvius, W.M. Howell, M. Kamali-Moghaddam, and O. Soderberg. 2010. Proximity ligation assays: a recent addition to the proteomics toolbox. Expert Rev Proteomics. 7:401-409.

Winkle, C.C., R.H. Olsen, H. Kim, S.S. Moy, J. Song, and S.L. Gupton. 2016. Trim9 Deletion Alters the Morphogenesis of Developing and Adult-Born Hippocampal Neurons and Impairs Spatial Learning and Memory. J. Neurosci. 36:49404958.

Yalgin, C., S. Ebrahimi, C. Delandre, L.F. Yoong, S. Akimoto, H. Tran, R. Amikura, R. Spokony, B. Torben-Nielsen, K.P. White, and A.W. Moore. 2015. Centrosomin represses dendrite branching by orienting microtubule nucleation. Nat. Neurosci. 18:1437-1445.